# Ethnobotany and Toxicity Status of Medicinal Plants with Cosmeceutical Relevance from Eastern Cape, South Africa

**DOI:** 10.3390/plants11111451

**Published:** 2022-05-30

**Authors:** Ashwell R. Ndhlala, Vuyisile S. Thibane, Cecilia M. Masehla, Phatlane W. Mokwala

**Affiliations:** 1Green Technologies Research Centre of Excellence, School of Agricultural and Environmental Sciences, University of Limpopo, Private Bag X1106, Sovenga 0727, South Africa; 2Department of Biochemistry and Biotechnology, Sefako Makgatho Health Sciences University, Ga-Rankuwa 0204, South Africa; vuyisile.thibane@smu.ac.za; 3Department of Biodiversity, University of Limpopo, Private Bag X1106, Sovenga 0727, South Africa; masehlacecilia@gmail.com (C.M.M.); phatlane.mokwala@ul.ac.za (P.W.M.)

**Keywords:** cosmeceutical, medicinal plants, natural products, skincare, toxicity

## Abstract

The indigenous people of the Eastern Cape residing within the richest plant biodiversity in the world, including Africa’s floral ‘gold mine’, have a long history of plant use for skincare. However, such rich flora comes with numerous plants that have the potential to cause harm to humans through their usage. Therefore, the study was aimed at documenting the toxicity status of important medicinal plants used by the indigenous people from the Eastern Cape for skincare and supported by literature for cosmeceutical relevance. A list of plants used for skincare was produced following an ethnobotanical survey. In addition, data on the level of toxicity and cosmeceutical relevance of plants listed from the survey were collected from literature resources. The study listed a total of 38 plants from 25 plant families, the majority being represented by the Asphodelaceae and Asteraceae, both at 13.2%. The most preferred plant parts were the leaves (60.4%) indicating sustainable harvesting practices by the community. The literature reports validated 70% of the medicinal plants surveyed for skincare were nontoxic. Most of the plants can be incorporated in the formulation of products intended for skincare due to their low toxicity and high cosmeceutical relevance.

## 1. Introduction

The Eastern Cape Province of South Africa is partly home to the Capensis Kingdom known as the Cape floral region. The region has a fynbos biome, endemic plant species, threatened plant species, and is one of the largest plant biodiverse regions in the world. The Cape floral region has eight protected areas from the Cape Peninsula right into the heart of the Eastern Cape province [1]. The large plant biodiversity in the province has encouraged the availability of numerous useful medicinal plants. Communities in the province have for a very long time lived an under-developed life with socio-economic challenges that have encouraged strong reliance and utilization of plants as a form of primary healthcare [2,3]. Plants are used for medication, cosmetics, herbs, spices, food, religion, and pasture [4].

The use of plants for wellbeing is a broad indication of a person’s quality of life and health. This holistic approach to human health can further be broken down into one’s care of their body, mind, and soul [5]. Medicinal plants have been known to maintain a person’s wellbeing by treating and preventing associated diseases [6,7]. People have always had the desire to look beautiful, and plants used in cosmetics are usually those that have phytochemical compounds with antioxidant properties. Beauty can be perceived as a state of good health both inside and outside [8,9]. However, some medicinal plants with intended effects can further have undesired side effects. These side effects are mostly undesirable and can be life-threatening to those using the plants. Babies are mostly susceptible to toxins because of few drug-metabolizing enzymes in their bodies. The patient’s response to plant toxins is influenced by age, genetic makeup, medical history, drug dosage, and other drug interactions [10]. The presence of toxins in the human blood system will also be characterized by changes in the skin. This is due to the skin and blood vessels interface, specifically around the face [11].

People in communities regard medicinal plants to be safe for use when they don’t show signs of short-term side effects such as gastrointestinal disturbances and body rush. In addition, long-term side effects such as cancer, liver, and kidney damage are mostly not associated with herbal use of medicinal plants. The absence of toxicity signs during the usage of medicinal plants depends on the method of analysis used. Toxicity lab analysis is the best method to determine the safety of medicinal plants [12]. Toxicity studies of medicinal plants involve the screening of cytotoxins, genotoxins, hepatoxins, nephrotoxins, neurotoxins, and environmental toxins [13,14,15]. Historically, the examination of medicinal plants in determining their safe dosages has been poorly documented. The lack of toxicity documentation can lead to prolonged usage of plants with inherent adverse effects for the user. Therefore, the aim of this study was to document the toxicity status of important medicinal plants with cosmeceutical relevance used by indigenous people from the Eastern Cape Province. The study further checked the toxicity status of documented plants with products and patents already formulated.

## 2. Materials and Methods

### 2.1. Study Area

The study area lies in the central part of the Eastern Cape Province and included nine villages from the Raymond Mhlaba Municipality. The distinct vegetation of the area in and around the study promotes plenty of plant biodiversity [16].

### 2.2. Ethnobotanical Survey

Participants with indigenous knowledge on the toxicity status of plants used for skincare were identified using purposive sampling method [17]. The interviews were conducted in the isiXhosa language using a semi-structured questionnaire with assistance from an interpreter fluent in both isiXhosa and English [18]. The data captured on the questionnaires was later translated to English. Where consent was given, audio and video assistance were used during the interview process. The interview process was designed to collect information on the demographics, names of plants used, type of plant, plant part used, method of preparation and administration, and reported toxicity or side effects. The data were used to calculate the frequency index to determine the most used plants using the equation; FI = (FC/N) × 100. Where, FC is the number of informants who reported on the plant’s toxicity status, and N is the total number of informants interviewed [19].

### 2.3. Data Collection

Plants mentioned in the survey were identified and voucher specimens were prepared and deposited at the Larry Leach Herbarium (UNIN), University of Limpopo. Data on the level of toxicity and product formulation of plants identified from the survey were collected from resources including journal articles, books, theses, and dissertations as well as databases, such as Google, Google Scholar, PubMed, ResearchGate, Science Direct, and Scopus. The search included but was not limited to the name of the identified plant together with phrases such as “toxicity”, “side effects” “formulation”, “product”, “prototype”, and “commercialization”.

### 2.4. Ethical Consideration

Ethical approval was obtained through the Turfloop Research Ethics Committee (TREC) at the University of Limpopo (UL) prior to commencement of the study. Further, permission from the office of the local chief was requested prior to the commencement of the survey. Participants signed a consent form prior to the interview process indicating their willingness to participate in the study. Personal information of informants was kept confidential on the questionnaires and was stored according to UL information storing policies. Agreement was achieved between the knowledge holders, community members, Raymond Mhlaba Economic Development Agency, and the researchers that any intellectual property and future commercial value emanating from the study will be shared equally amongst the interested parties.

## 3. Results

### 3.1. Ethnobotanical Survey

A total number of 50 informants were interviewed, with the majority of the knowledge holders being female at 60.6% and males at 38.4%, indicative that women are the caretakers of indigenous knowledge [20]. The indigenous knowledge holders were aged between 21 and 80 years, with those within the age range of 21–40 accounting 6.0%, while those aged 41–60 years accounting 45.5%, and 61–80 years accounting 48.5% (Figure 1).

Figure 2 shows the different plant parts reported in the survey and used traditionally for skincare. The most preferred plant parts were the leaves (60.4%), indicating that the communities are very much aware of sustainable harvesting techniques. The second highest plant part used was the bulb at 13.2%. Challenges with the use of bulbs is the destruction of the plants associated with their use. The use of roots (9.4%), leaf gel (5.7%), bark (3.8%), and seeds (3.8%) were significantly high when compared to the stem and leaf sap, with their use reported at 1.9% [21].

The method of preparation varied between the plants with the decoction method at 44.8% being the most preferred (Figure 3). Application of the plants as a paste was the second-highest at 14% usage. This would be within an acceptable range as many of the plants were used for skincare. The use of plants prepared through infusion and cooking was significantly high with a recording of 10.3% and 8.6%, respectively. Plant preparation methods were further recorded for raw leaves, ground leaves, plant oil sap, bathing, and steaming all at 1.7%.

### 3.2. Toxicity Status

The frequency index (FI) as expressed in Table 1 was relatively high for *Artemisia afra* (34.32), *Bulbine frutescens* (18.72), *Aloe ferox* (18.72), *Alepidea amatymbica* (15.60), *Clausena anisata* (15.60), *Hypoxis hemerocallidea* (15.60), *Persea americana* (15.60), *Cassipourea flanaganii* (12.48), *Helichrysum petiolare* (12.48), *Marrubium vulgare* (12.48), *Ilex mitis* (12.48), *Haemanthus albiflos* (9.36), and *Scabiosa albanensis* (9.36) when compared to the other plants. The results show the use of the plants being distributed over a range of plant families. These can partly be attributed to the fact that skincare entails a more holistic approach on skin health.

Table 1 provides the toxicity status reported from the survey of plants used by the Xhosa communities in the Raymond Mhlaba Municipality for skincare. A total number of 38 plants was recorded from the survey. The highest number of plant families represented were from the Asphodelaceae and Asteraceae, both at 13.2%. These were followed by the Amaranthaceae, Amyrallidaceae, Brassicaceae, and Rutaceae with all represented at 5.3%. The Anacardiaceae, Apiaceae, Apocynaceae, Aquifoliaceae, Balanophoraceae, Boraginaceae, Cannabaceae, Caprifoliaceae, Caricaceae, Euphorbiaceae, Gunneraceae, Hyacinthaceae, Hypoxidaceae, Lamiaceae, Lauraceae, Moringaceae, Plantaginaceae, Polygonaceae, Rhizophoraceae, and Urticaceae with each family represented at 2.6%. The toxicity or side effects of the plants reported from the survey were compared with literature, specifically looking at the reported toxicity status, assay method, lethal dosage, and plant parts used. Table 2 represents patents for products already in the market that were formulated from plants highlighted to be used for skincare from our survey. The toxic effect associated with medicinal plants is one of the challenging territories to charter in natural product research as many of the knowledge holders believe plants not to be toxic. The validation of the efficacy of some of the reported claims of medicinal plants is not to dispute the indigenous knowledge but rather to ensure the safe usage of many of these plants. Plants are known to produce toxins as defense against herbivores, microorganisms, and viruses. These toxins have further been reported to be harmful to humans who consume or use them [22].

**Table 1 plants-11-01451-t001:** Toxicity status for plants with cosmeceutical relevance used by Xhosa communities in the Raymond Mhlaba Municipality, Amathole district, Eastern Cape Province.

No.	Scientific Name	Voucher Number	Common Names	FI	Recorded Toxicity from Survey	Reported Toxicity from Literature	Reference
1	Amaranthaceae *Amaranthus hybridus* L.	M.C.12	Utyuthu (Xh) Green amaranth (Eng)	3.12	No	Non cytotoxic Brine shrimp assay LC_50_: 6233.6 µg/mL Leaves	[23]
2	Amaranthaceae *Chenopodium album* L.	M.C.44	Imbikicane (Xh) Lamb’s quarters (Eng)	3.12	No	Nontoxic Mortality rate assay LD_50_: 71.3 mg/kg Leaves	[24]
3	Amaryllidaceae *Allium sativum* L.	M.C.32	Ivimbampunzi (Xh) Garlic (Eng)	6.24	No	Nontoxic Mortality rate assay LD_50_: 3034 mg/kg Bulb	[25]
4	Amaryllidaceae *Haemanthus albiflos* Jacq.	M.C.33	Umathunga (Xh)	9.36	No	Cytotoxic NIH 3T3 cell line LD_50_: 3.24 mg/mL Bulb	[26]
5	Anacardiaceae *Schinus molle* L.	M.C.46	Peperibhomu (Xh) Peruvian pepper (Eng)	3.12	No	Cytotoxic K562 cell line LD_50_: 78.70 µg/mL Essential oil	[27]
6	Apiaceae *Alepidea amatymbica* Eckl. & Zeyh.	M.C.29	Iqwili (Xh) Larger tinsel flower (Eng)	15.60	No	Non genotoxic Ames test Number of His+ revertant Rhizome	[28]
7	Apocynaceae *Acokanthera oblongifolia* (Hochst.) Codd	M.C.09	Ubuhlungu (Xh) Dune poison (Eng)	3.12	No	Non mutagenic Ames test Number of His+ revertant Twigs	[29]
8	Aquifoliaceae *Ilex mitis* (L.) Radlk.	M.C.21	Isidumo (Xh) Cape holy (Eng)	12.48	No	No toxicity study	No reports
9	Asphodelaceae *Aloiampelos ciliaris* Haw. Klopper & Gideon F.Sm. var. *ciliaris*	M.C.01	Ikhala (Xh) Climbing aloe (Eng)	3.12	Laxative effect	No toxicity study	No reports
10	Asphodelaceae *Aloe ferox* Mill.	M.C.05	Ikhala (Xh) Bitter aloe (Eng)	18.72	Laxative effect	Nontoxic in vivo assay (mice) LD_50_: >5.0 g/kg Resin extract	[30]
11	Asphodelaceae *Aloiampelos tenuior* (Haw.) Klopper & Gideon F.Sm.	M.C.04	Impapane (Xh) Slender aloe (Eng)	3.12	No	No toxicity study	No reports
12	Asphodelaceae *Bulbine abyssinica* A. Rich.	M.C.06	Uyakayakana (Xh) Geelkatstert (Afk)	6.24	No	Nontoxic Brine shrimp assay LD_50_: 3120 µg/mL Oil	[31]
13	Asphodelaceae *Bulbine frutescens* (L.) Willd.	M.C.07	Itswela le nyoka (Xh) Balsem kopieva (Afr)	18.72	No	Cytotoxic Chang liver cells LD_50_: 62.5 µg/mL Whole plant	[32]
14	Asteraceae *Arctotis arctotoides* (L.f.) O.Hoffm	M.C.50	Ubushwa (Xh) Botterblom (Afr)	3.12	No	Cytotoxic Brine Shrimp assay LD_50_: 1000 µg/mL Leaves	[16]
15	Asteraceae *Sonchus asper* (L.) Hill	M.C.14	Ihlaba (Xh) Spiny sowthistle (Eng)	3.12	No	Nontoxic in vivo assay (rats) LD_50_: 200 mg/kg Whole plant	[33]
16	Asteraceae *Artemisia afra* Jacq. ex Willd.	M.C.02	Umhlonyana (Xh) Wind wormwood (Eng)	34.32	No	Nontoxic in vivo assay (mice) LD_50_: 9833.4 mg/kg Leaves	[34]
17	Asteraceae *Helichrysum petiolare* Hilliard & B.L.Burtt	M.C.40	Impepho (Xh) Kooigoed (Afr)	12.48	No	Non mutagenic Ames test Number of His+ revertant Leaves	[35]
18	Asteraceae *Senecio inornatus* DC.	M.C.08	Inkanga (Xh) Tall marsh senecio (Eng)	6.24	No	No toxicity study	No reports
19	Balanophoraceae *Sarcophyte sanguinea* Sparrm. subsp. *sanguinea*	M.C.37	Umavumbuka (Xh) Wolwekos (Afr)	3.12	No	Non cytotoxic Monkey kidney cell line LD_50_: 50 µg/mL Stem bark	[36]
20	Boraginaceae *Symphytum officinale* L.	M.C.31	Izicwe (Xh) Comfrey (Eng)	3.12	No	Genotoxic Liver *cll* gene mutations Leaves	[37]
21	Brassicaceae *Brassica oleracea* L.	M.C.20	Kale (Eng)	3.12	No	Nontoxic in vivo assay (rats) LD_50_: 5000 mg/kg Leaves	[38]
22	Brassicaceae *Rorippa nasturtium-aquaticum* (L.) Hayek	M.C.15	Uwatala (Xh) Watercress (Eng)	3.12	No	Nontoxic in vivo assay (rats) LD_50_: 500 mg/kg Leaves	[39]
23	Cannabaceae *Cannabis sativa* L.	M.C.38	Umya (Xh) Hemp (Eng)	3.12	No	Cytotoxic Brine shrimp assay LD_50_: 13.6 µg/mL Oil	[40]
24	Caprifoliaceae *Scabiosa albanensis* R.A.Dyer	M.C.36	Isilawu (Xh) Scabious (Eng)	9.36	No	No toxicity study	No reports
25	Caricaceae *Carica papaya* L.	M.C.30	Ipopo (Xh) Paw Paw (Eng)	6.24	No	Cytotoxic in vivo assay (catfish) LC_50_: 1.29 mg/mL Seeds	[41]
26	Euphorbiaceae *Acalypha glabrata* Thunb.	M.C.25	Umthombothi (Xh) Forest false nettle (Eng)	6.24	No	No toxicity study	No reports
27	Gunneraceae *Gunnera perpensa* L.	M.C.41	Iphuzi (Xh) River pumpkin (Eng)	3.12	No	Nontoxic in vivo assay (rats) LD_50_: 400 mg/kg Leaves	[42]
28	Hyacinthaceae *Albuca setosa* Jacq.	M.C.39	Inqwebeba (Xh) Small white (Eng)	6.24	No	Cytotoxic MDA-MB-231 breast cancer cell line 64.52 µg/mL Bulb	[43]
29	Hypoxidaceae *Hypoxis hemerocallidea* Fisch., C.A. Mey. and Ave-Lall.	M.C.34	Inongwe (Xh) Yellow star (Eng)	15.60	No	Nontoxic Monkey Vero kidney cell line LD_50_: 95.51 µg/mL Bulb	[44]
30	Lamiaceae *Marrubium vulgare* L.	M.C.14	Umhlonyane (Xh) Horehound (Eng)	12.48	No	Cytotoxic Brine shrimp assay LC_50_: 112.65 µg/mL Leaves	[45]
31	Lauraceae *Persea americana* Mill.	M.C.19	Iavokado (Xh) Avocado Tree (Eng)	15.60	No	Nontoxic in vivo assay (mice) LD_50_: >4000 mg/kg Seeds	[46]
32	Moringaceae *Moringa oleifera* Lam.	M.C.35	Moringa (Eng) Peperwortelboom (Afr)	3.12	No	Nontoxic in vivo assay (rats) In vivo assay LD_50_: 3000 mg/kg Leaves	[47]
33	Plantaginaceae *Plantago lanceolata* L.	M.C.13	Ubendlela (Xh) Narrowleaf plantain (Eng)	3.12	No	Nontoxic in vivo assay (mice) LD_50_: 12 mL/kg Leaves syrup	[48]
34	Polygonaceae *Emex australis* Steinh.	M.C.43	Inkunzane (Xh) Souther three corner jack (Eng)	3.12	No	No toxicity study	No reports
35	Rhizophoraceae *Cassipourea flanaganii* (Schinz) Alston.	M.C.18	UmMemezi (Xh)	12.48	No	Cytotoxic HEM cell line LD_50_: 100 µg/mL Bark	[49]
36	Rutaceae *Clausena anisata* (Willd.) Hook.f. ex Benth.	M.C.11	Iperipes (Xh) Horsewood (Eng)	15.60	No	No toxicity study	No reports
37	Rutaceae *Ruta graveolens* L.	M.C.45	Ivendrithi (Xh) Rue (Eng)	3.12	No	Cytotoxic in vivo assay (mice) LD_50_: <1000 mg/kg Leaves	[50]
38	Urticaceae *Urtica urens* L.	M.C.03	Uralijan (Xh)	3.12	No	Nontoxic in vivo assay (rats) LD_50_: >5000 mg/kg Leaves	[51]

Xh-Xhosa; Eng-English; Afr-Afrikaan; FI-Frequency index.

**Table 2 plants-11-01451-t002:** Patents/products on medicinal plants with cosmeceutical relevance.

No.	Medicinal Plant	Product Description	Application	Dosage	Patent Number	Reference
1	*Schinus molle* L.	Hydroethanolic plant extract prepared from aerial parts of the plant. Comprised mainly of quercitrin (0.04%) and miquelianin (0.02%). Cosmetic composition used as protective active agent and to improve barrier function of the skin.	Topical application	2 mg/cm^2^	US11045669B2	[52]
2	*Cannabis sativa* L.	A topical formulation used in treating dermatological diseases, comprising a Cannabis derived botanical drug product, wherein the concentration of tetrahydrocannabinol, cannabidiol, or both in the topical formulation is greater than 2 mg/kg.	Topical application	Not specified	US010226496B2	[53]
3	*Bulbine frutescens* L.	A medicament related to treatment of damaged skin because of scarring, aging, and excessive exposure to UV light. The topical application comprises of *B. frutescens* (9.9–11% *m*/*m*) and *Centella asiatica* (0.45–0.55% *m*/*m*), supplemented by oleuropein (0.18–0.3% *m*/*v*).	Topical application	Microporous tape with topical scar gel	USO08071139B2	[54]
4	*Symphytum officinale* L.	An injectable solution (minimal dose for intramuscular administration to an adult is of 30 mg/day/70 kg body weight) of *S. officinale* prepared from freshly cut roots, with anti-inflammatory effect.	Injectable syringe	5 mL ampoule	USOO7604822B2	[55]
5	*Ruta graveolens* L.	Hydroalcoholic plant extract prepared from leaves used to prevent and/or treat arterial hypertension, as it exhibits a marked vasodilator effect.	Oral administration	Not specified	WO201407277Al	[56]

## 4. Discussion

The reported literature was able to validate 70% of the medicinal plants surveyed for skincare use in the study to being nontoxic. Some of the popularly used medicinal plants were reported to express some degree of toxicity. Fresh leaf juice of *B. frutescens* has largely been reported for use in skincare and wound healing [57]. A whole-plant extract of *B. frutescens* was reported to exhibit acute cytotoxic effect (LD_50_ < 1000 µg/mL) on Chang liver cells [32]. It is important to further note that some of the reported side effects in literature exhibit the intended use of these plants. The intended recreational use of *C. sativa* can largely be attributed to the plant’s acute cannabis intoxication [58,59]. This was further evident by the reported laxative effect of *A. ciliaris* and *A. ferox* from the Asphodelaceae family. However, some of the intended use of the plant can have adverse effects with prolonged usage. The use of *C. flanaganii* has been reported to enhance skin beauty and complexion [20]. However, the apparent mode of action was because of the reduction of the total number of melanocytes due to the plant’s toxic effect [49]. The highest FI of 34.4 in the current study was reported to be *A. afra* with the plant reported to relieve the body of coughs and colds. However, the complex volatile compounds, such as thujone found in the oil of the plant, have previously been reported to cause confusion, convulsions, and ending in a coma with some patients due to high doses [60]. Even though the informants reported the medicinal plants to be nontoxic, extremely high dosages can be lethal and express long-term effects on human health as is with all other plants. There were no toxicity studies that could be found in the literature for *Ilex mitis*, *A. ciliaris*, *A. tenuior*, *S. inornatus*, *S. sanguinea*, *S. albanensis*, *A. glabrata*, *E. australis,* and *C. anisata*. Future studies on the toxicity or side effects associated with these important medicinal plants will be undertaken to lift the lid on their safety and prospects as used traditionally in skincare.

The need for the commercialization of South African indigenous plants has been expressed by several researchers [61,62]. The successful commercialization of these plants will aid in alleviating some of the socio-economic challenges faced by poor communities with indigenous knowledge on plants usage. These communities can benefit through job creation and bioprospecting agreements related to some of these plants. Toxicity studies play a significant role in product formulation. The data presented in Table 2 indicate that many of these plants are still not yet commercialized. Only 13% of the plants reported in the survey to be used for skincare have products that are patented. However, it is interesting to note that even though the plants have products that are patented, reports have indicated these plants to either be cytotoxic or genotoxic. Leaves of *S. officinale* were reported to express genotoxic traits in enabling liver *cll* gene mutations [37]. However, roots of *S. officinale* were successfully incorporated into an injectable anti-inflammatory product [55]. A whole plant preparation of *B. frutescens* was also reported to express cytotoxic traits on Chang liver cells [32]. However, a topical medicament prepared with 9.9–11% *m*/*m* of *B. frutescens* leaves extract used against damaged skin has been patented [54]. These studies demonstrated that different plant parts and varying levels and, combinations of the extract used, can express variations in toxicity. Similarly, extracts of *S. mole*, *C. sativa,* and *R. graveolens* that were reported to express some degree of toxicity have been successfully incorporated into products as reported on their patents. This report opens opportunities for commercialization, especially with those neglected plants that were reported to exhibit toxic and nontoxic traits.

## 5. Conclusions

The indigenous knowledge on the use of plants is still possessed by the elderly in the community as the study has reported. However, the report on informants as young as 21 years is a promising outlook to transfer indigenous knowledge from the elderly to the young. The study has further reported on the safe use of the majority of the surveyed plants due to their perceived low toxicity.

## Figures and Tables

**Figure 1 plants-11-01451-f001:**
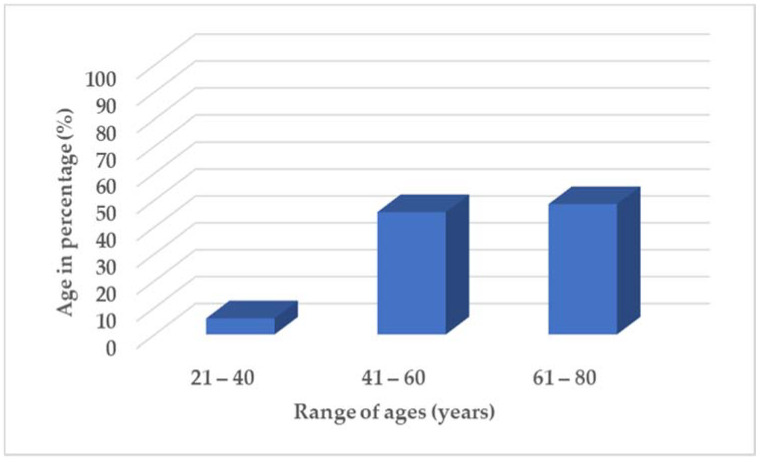
Age range in percentage (%) of knowledge holders participated in survey.

**Figure 2 plants-11-01451-f002:**
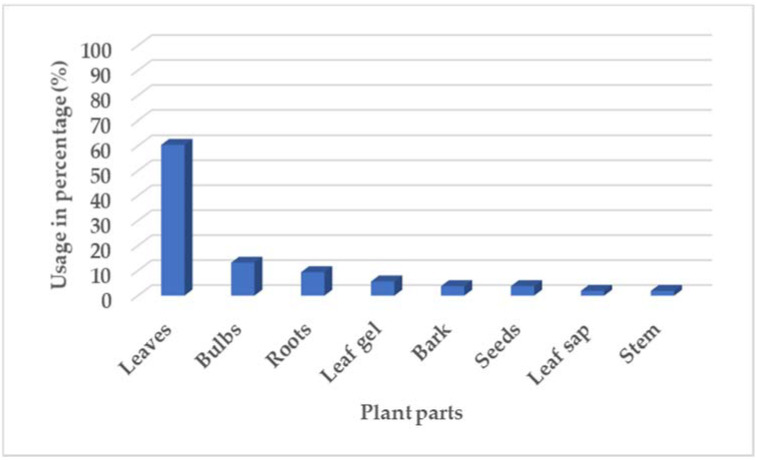
Distribution of plant parts in percentage (%) reported in survey.

**Figure 3 plants-11-01451-f003:**
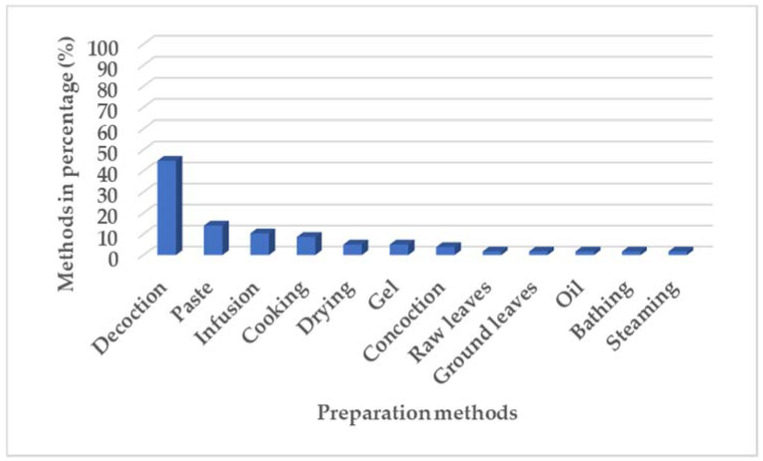
Preparation methods in percentage (%) reported in survey.

## Data Availability

Not applicable.

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
