# Peer review of "Ethnobotany and Toxicity Status of Medicinal Plants with Cosmeceutical Relevance from Eastern Cape, South Africa"

_plants, 2022, doi:10.3390/plants11111451_

Round 1
Reviewer 1 Report
This review paper is very interesting because it is a good contribution for the knowledge about the ethnobotany and toxicity status of medicinal plants with cosmeceutical relevance from Eastern Cape (South Africa).
The study was aimed at documenting the toxicity status of important medicinal plants used by the indigenous people from the Eastern Cape for skincare and supported by literature for cosmeceutical relevance.
The article has three Figures, two Tables and presents an interesting study well discussed and which summarized a huge amount of information from important electronic databases and other sources. The conclusion was supported by the bibliographic references.
There are some minor changes that need to be made but which are of the presentation nature. The proposed review changes are as follows:
- On page 6 (Table 1, Nº 31), “Persea Americana Mill.” should be “Persea americana Mill.”;
- On lines 137-138, “Cannabis sativa” should be in italic.
- I would like to suggest for the structure/position of some of the sections, so that they come in the order and numbering as follows:
- Point 4, “Materials and Methods” should become Point 2 (with all subsections updated accordingly: 4.1 -> 2.1, 4.2 -> 2.2, 4.3 -> 2.3, 4.4 -> 2.4)
- Point 2, “Results” should become Point 3 (with all subsections updated accordingly: 2.1 -> 3.1, 2.2 -> 3.2)
- Point 3, “Discussion” should become Point 4 and should be named “Discussion and Conclusions”.
With these minor changes, the recommendation will be to accept the manuscript for publication.
Author Response
Reviewer 1
On page 6 (Table 1, Nº 31), “Persea Americana Mill.” should be “Persea americana Mill
Response: The name has been amended.
On lines 137-138, “Cannabis sativa” should be in italic.
Response: The name has been written in italics
I would like to suggest for the structure/position of some of the sections, so that they come in the order and numbering as follows:
- Point 4, “Materials and Methods” should become Point 2 (with all subsections updated accordingly: 4.1 -> 2.1, 4.2 -> 2.2, 4.3 -> 2.3, 4.4 -> 2.4)
- Point 2, “Results” should become Point 3 (with all subsections updated accordingly: 2.1 -> 3.1, 2.2 -> 3.2)
- Point 3, “Discussion” should become Point 4 and should be named “Discussion and Conclusions”.
Response: The structure has been rearranged as per reviewer suggestion
Reviewer 2 Report
Criticism
- From the data presented in Table 1, it is not clear how the “Method of preparation” is related to cosmeceutical use? How are “cough relief”, “flu treatment”, “relieve of coughs and headaches”, “relief from stomach aches” related to cosmeceutical properties?
- The data in Table 1 does not allow the "Method of preparation" to be associated with the toxicity of the obtained extract. How, for example, decoction from leaves Schinus molle L. is associated with toxicity (cytotoxic) of essential oil [Díaz, C.; Quesada, S.; Brenes, O.; Aguilar, G.; Cicció, J.F. Chemical Composition of Schinus Molle Essential Oil and Its Cytotoxic Activity on Tumour Cell Lines. Natural Product Research 2008, 22, 1521–1534, doi:10.1080/14786410701848154]. Can paste prepared from Symphytum officinale leaves applied topically for wound treatment cause Genotoxic Liver cll gene mutations?
- The authors correctly point out that «These studies demonstrated that different plant parts and varying levels and, combinations of the extract used can express variations in toxicity». However, data on doses and frequency of use of drugs are not available.
- There are inaccuracies in the Latin names of plants in tables 1 and 2. The authors need to indicate on the basis of what sources the Latin names of plants were verified.
- Section Conclusion in the manuscript is missing.
Author Response
Reviewer 2
From the data presented in Table 1, it is not clear how the “Method of preparation” is related to cosmeceutical use? How are “cough relief”, “flu treatment”, “relieve of coughs and headaches”, “relief from stomach aches” related to cosmeceutical properties?
Response: Thank you for the comment. Studies have previously reported on how the method of preparation and the different plant parts used affect the toxicity status of the plant, hence the authors sort to also present the data in the current study. However, the table has been updated and the method of preparation excluded.
The data in Table 1 does not allow the "Method of preparation" to be associated with the toxicity of the obtained extract. How, for example, decoction from leaves Schinus molle L. is associated with toxicity (cytotoxic) of essential oil [Díaz, C.; Quesada, S.; Brenes, O.; Aguilar, G.; Cicció, J.F. Chemical Composition of Schinus Molle Essential Oil and Its Cytotoxic Activity on Tumour Cell Lines. Natural Product Research 2008, 22, 1521–1534, doi:10.1080/14786410701848154]. Can paste prepared from Symphytum officinale leaves applied topically for wound treatment cause Genotoxic Liver cll gene mutations?
Response: Thank you for the comment. The reported toxicity from literature presented in the current study is intended to compare the indigenous knowledge on plant toxicity as reported by the survey with the generated scientific data on the toxicity status of similar plants. However, the table has been updated and the method of preparation excluded.
The authors correctly point out that «These studies demonstrated that different plant parts and varying levels and, combinations of the extract used can express variations in toxicity». However, data on doses and frequency of use of drugs are not available.
Response: Table updated with doses and frequency applications.
There are inaccuracies in the Latin names of plants in tables 1 and 2. The authors need to indicate on the basis of what sources the Latin names of plants were verified.
Response: Thank you for the comment. Inaccuracies were detected and corrected using national biodiversity databases of PlantZAfrica (hosted by the South African National Biodiversity Institute) and the Royal Botanic Gardens (Kew).
Section Conclusion in the manuscript is missing.
Response: Conclusions have been added.
Round 2
Reviewer 2 Report
The authors' answers are clear. There are no comments.